# Inhibition of Bromodomain Proteins Enhances Oncolytic HAdVC5 Replication and Efficacy in Pancreatic Ductal Adenocarcinoma (PDAC) Models

**DOI:** 10.3390/ijms25021265

**Published:** 2024-01-19

**Authors:** Tizong Miao, Alistair Symonds, Oliver J. Hickman, Dongsheng Wu, Ping Wang, Nick Lemoine, Yaohe Wang, Spiros Linardopoulos, Gunnel Halldén

**Affiliations:** 1Centre for Biomarkers and Biotherapeutics, Barts Cancer Institute, Queen Mary University of London, London EC1M 6BQ, UK; t.miao@qmul.ac.uk (T.M.); bci-director@qmul.ac.uk (N.L.); yaohe.wang@qmul.ac.uk (Y.W.); 2Centre for Immunobiology, Blizard Institute, Barts and The London School of Medicine and Dentistry, Queen Mary University of London, London E1 2AT, UK; a.l.j.symonds@qmul.ac.uk (A.S.); p.wang@qmul.ac.uk (P.W.); 3Cancer Drug Target Discovery Laboratory, The Institute of Cancer Research, London SW3 6JB, UK; oliver.hickman@icr.ac.uk (O.J.H.); spiros.linardopoulos@icr.ac.uk (S.L.); 4Bioimaging Centre, School of Engineering and Materials Science, Queen Mary University of London, London E1 4NS, UK; dong.wu@qmul.ac.uk

**Keywords:** BRD4 inhibitors, small molecule screen, oncolytic adenovirus, iBET-762, OTX-015, Neo-273, organoid cultures, co-cultures, stromal cells, in vivo

## Abstract

Pancreatic ductal adenocarcinoma (PDAC) is the most aggressive type of pancreatic cancer, which rapidly develops resistance to the current standard of care. Several oncolytic Human AdenoViruses (HAdVs) have been reported to re-sensitize drug-resistant cancer cells and in combination with chemotherapeutics attenuate solid tumour growth. Obstacles preventing greater clinical success are rapid hepatic elimination and limited viral replication and spread within the tumour microenvironment. We hypothesised that higher intratumoural levels of the virus could be achieved by altering cellular epigenetic regulation. Here we report on the screening of an enriched epigenetics small molecule library and validation of six compounds that increased viral gene expression and replication. The greatest effects were observed with three epigenetic inhibitors targeting bromodomain (BRD)-containing proteins. Specifically, BRD4 inhibitors enhanced the efficacy of Ad5 wild type, Ad∆∆, and Ad-3∆-A20T in 3-dimensional co-culture models of PDAC and in vivo xenografts. RNAseq analysis demonstrated that the inhibitors increased viral E1A expression, altered expression of cell cycle regulators and inflammatory factors, and attenuated expression levels of tumour cell oncogenes such as c-Myc and Myb. The data suggest that the tumour-selective Ad∆∆ and Ad-3∆-A20T combined with epigenetic inhibitors is a novel strategy for the treatment of PDAC by eliminating both cancer and associated stromal cells to pave the way for immune cell access even after systemic delivery of the virus.

## 1. Introduction

Pancreatic ductal adenocarcinoma (PDAC) is a cancer of unmet clinical need and has the lowest 5-year survival rates of all cancers globally (<5%) [1]. The major reasons for the poor survival rates are the late presentation of symptoms and the rapid development of resistance to all current standards of care including the cytotoxic drug gemcitabine [2,3]. Without novel curative therapies, most patients will suffer from severe morbidity and succumb to the disease despite recent progress in early biomarker identification [4,5]. Novel therapies that negate drug resistance are urgently needed. Oncolytic adenoviral (OAd) mutants are promising candidates that selectively replicate in and kill cancer cells, including treatment-resistant cells. Powerful and specific cancer cell killing via cell lysis releases novel tumour antigens, cellular pathogen-associated and damage-associated molecular patterns (PAMPs and DAMPS), that promote activation of the host anti-tumour immune responses [6,7]. Several OAd mutants have been evaluated in clinical trials targeting a variety of solid tumours including PDAC [8,9] or are currently in early phase trials (e.g., NCT02045589; NCT02045602; NCT02705196). Overall safety has been demonstrated in thousands of patients while efficacy was limited. Contributing factors to the moderate clinical results in PDAC include poor viral uptake and spread within a dense tumour microenvironment (TME) and attenuated viral replication in combination with standard of care therapies [10,11,12,13,14].

To address the limited viral replication and spread in the TME, numerous OAd mutants have been engineered to target tumour-specific antigens or express ECM-degrading enzymes [15,16]. We previously engineered the cancer-optimised Ad∆∆ and Ad-3∆-A20T, which are highly cancer-selective and efficacious in models of PDAC and synergise with cytotoxic drugs including gemcitabine [13,14,17,18]. Selectivity and potency are caused by the deletions of the E1ACR2-domain (pRb-binding) and the anti-apoptotic Bcl2-homologue E1B19K, exploiting the typical alterations in late-stage PDAC such as de-regulation of the pRb/p16-p53 and apoptosis pathways and KRAS-activating mutations [13,17,18]. A third deletion of the MHC-binding protein E3gp19K re-activates the host immune responses towards the infected cancer cells by enabling efficient MHC class I-antigen presentation [19]. Ad∆∆ infects epithelial cells including adenocarcinomas by binding the viral fibre protein to the Coxsackievirus and Adenovirus Receptor (CAR) followed by penton-binding to αvβ3- and αvβ5-integrins. The virus is subsequently internalized via the endosome, and the viral DNA is transported to the nucleus for gene transcription. The Ad-3∆-A20T mutant has an additional alteration to improve bioavailability and decrease uptake in normal epithelial cells. Ad-3∆-A20T is de-targeted from CAR binding and re-targeted to αvβ6-integrin-expressing tumours by insertion of a 20-amino acid integrin-ligand from Foot-and-Mouth-Disease-Virus (FMDV) in the fibre knob [14,20]. The avβ6-integrins are highly expressed in PDAC and exclusively expressed in many solid cancers [21,22]. Both mutants are highly efficacious, selective, and non-toxic in preclinical models of PDAC including cultured cells and human tumour xenografts in athymic mice, and promote cell killing in combination with cytotoxic drugs in treatment-resistant cells.

Synergistic cell killing was observed with Ad∆∆ and Ad-3∆-A20T in combination with DNA-damaging drugs; however, viral replication was attenuated, preventing efficient spread within the TME [13,14]. To further optimize the potency of Ad∆∆ and Ad-3∆-A20T, we hypothesized that combinations with molecules that do not cause DNA damage or induce apoptosis would be more beneficial for viral replication. It is well established that HAdVC5 infection remodels the cellular epigenetic landscape by mediating deacetylation of differentiation-related genes and re-acetylation of promoters controlling processes that are requirements for adenovirus propagation [23,24,25,26,27]. These complex and interdependent functions are exemplified by E1A recruitment of the p300/CBP histone acetyltransferases during early infection, which removes the complex from active promoters and relocates the complex to promoters benefiting viral functions [26]. In previous studies we demonstrated that histone deacetylase (HDAC) inhibitors such as TSA, MS-275, Romidepsin, and Scriptaid promoted viral propagation in breast cancer and PDAC models [28]. To build on these findings and extend our studies to novel inhibitors, we screened an enriched epigenetics small molecule compound library (181 molecules) to identify molecules that increased viral replication in PDAC models. We identified and validated six compounds that increased early gene expression and replication. Three highly potent bromodomain-4 (BRD4) and p300/CBP inhibitors, OTX-015, iBet-762, and Neo-2734, were selected for further in-depth studies. Bromodomains (~110 amino acids) are present in numerous regulatory proteins and are essential for association with acetylated lysine residues on N-terminal histone tails. This binding mediates chromatin remodelling, in turn facilitating active gene transcription. Members of the bromodomain-containing protein family are the Bromo- and Extra-Terminal domain (BET) proteins including BRD4 which is part of the transcription complex regulating c-Myc expression.

We demonstrated that the inhibitors greatly decreased c-Myc expression and boosted viral replication and oncolysis in PDAC cells and cancer-associated fibroblasts (CAFs) in 3-dimensional co-cultures (3D) and in vivo murine xenograft models. Data from our RNAseq analysis showed that factors essential for viral protein and DNA synthesis were upregulated and host cell immune-related pathways were downregulated by the inhibitors, while viral early gene expression was greatly increased. From these findings, we conclude that the specific inhibition of BRD4-dependent transcription is beneficial for viral propagation and anti-tumour efficacy and is a promising avenue for future clinical translation.

## 2. Results

### 2.1. Optimization of Conditions for High-Throughput Screening (HTS)

Conditions were established to screen a small molecule epigenetic compound library for increased viral replication (Figure 1A). Culture conditions for the PDAC-representative Suit-2 cells expressing the typical activating KRAS mutation (KRAS G12D), *CDKN2A/p16* gene deletion, and inactivating TP53 mutations [29] were optimized. The characteristics of Suit-2 cells enabled growth in 384-well format and infection with Ad5wtGFP at increasing doses and time. Expression of GFP was used as a marker for viral uptake (early gene expression) and replication in the presence of test compounds known to increase and decrease viral uptake and replication, Lysine and Gemcitabine, respectively (Figure 1B; Appendix A). Lysine enhanced GFP expression after 24 h and viral genome copies after 48 h due to the higher number of particles infecting the cells. In contrast, Gemcitabine decreased GFP expression and viral genome amplification at both time points. Importantly, these data proved that the experimental conditions supported the detection of changes in viral uptake and replication and were suitable for HTS.

### 2.2. High Throughput Screening (HTS) of a Small Molecule Compound Library

Each compound in the library was tested at 10 nM, 100 nM, 500 nM, 1 µM, and 5 µM in triplicate samples for analysis of changes in viral GFP expression (E1A-driven expression) at 24 h post-infection. Integrated mean GFP values and Z-scores were determined for GFP+ live cells in each condition (representative results for compounds added at 0.5 µM; Figure 1C). Combining the results from screens at all tested dose levels (1000 screened combinations) indicated 50–60 GFP-positive hits. Viral genome copies were determined by qPCR analysis of the same wells to determine increases in replication (selected wells shown; Figure 1D). From the genome quantification, six compounds were verified to act as potential enhancers of viral replication; Barasertib (0.5 µM; Aurora B inhibitor), CPI-360 (5 µM; EZH2 inhibitor), MLN8054 (100 nM; Aurora A inhibitor), iBet-762 (0.5 µM; Bromodomain inhibitor), OTX-015 (0.5 µM; Bromodomain inhibitor), and GSK2801 (1 µM; BAZ2B/A bromodomain inhibitor). The six drugs were validated in independent infection experiments by qPCR (Figure 1E). Three compounds showed consistent increases in viral replication, including CPI-360, iBET-762, and OTX-015. The more significant increases were detected at all tested doses with iBET-762 and OTX-015. Barasertib, GSK2801, and MLN8054 needed further dose and time optimisation and were not pursued in this study. Both iBET-762 and OTX-015 belong to the bromodomain and extra-terminal motif (BET) protein inhibitors that block binding of bromodomain 4 (BRD4)-containing coactivators to acetylated transcription factors, thereby preventing transcription of target genes [30,31,32]. For example, BRD4-containing proteins play major roles as coactivators of c-Myc expression, which is frequently upregulated in PDAC, and support cancer cell survival and proliferation [33,34]. Therefore iBET-762 and OTX-015 were selected for further in-depth studies.

### 2.3. The Bromodomain Inhibitors OTX-015 and iBet-762 Promote Virus Replication Rather Than Uptake

To explore whether OTX-015 and iBet-762 increased cellular uptake of virus and thereby increased replication, Ad5wtGFP-infected and inhibitor-treated Suit-2 cells were analyzed by flow cytometry for early GFP expression prior to replication. Limited effects on the number of infected cells (% GFP) were observed in the presence of iBet-762 or OTX-015 (Figure 2A; left). The polybrene positive control increased viral uptake from 9% (DMSO) to 30%, while the inhibitors increased viral uptake to only 11–14%. In contrast, GFP-expression/cell (MFI) increased from 17,000 up to 33,000 with the inhibitors and 17,000 to 24,000 with the polybrene control, indicating that the BRD4 inhibitors selectively increased replication within cells rather than uptake of the virus (Figure 2A; right). In agreement with these findings, GFP-expression/cell was notably increased at the later time points in cells treated with the inhibitors (98,000 and 112,000; 72 h post-infection) compared to untreated and polybrene controls (59,000) (Figure 2B; right). The results demonstrate that iBet-762 and OTX-015 do not significantly affect viral uptake as determined by % of GFP-expressing cells after 24–48 h (Figure 2B; left). Minor increases were noted after 48 and 72 h, indicating improved viral spread (replication) and new rounds of infection, demonstrated by the greater increases in MFI at 72 h in the presence of the inhibitors.

Significant drug-mediated increases in genome copy numbers at these time points with both Ad∆∆ in Suit-2 cells and Ad-3∆-A20T in Panc04.03 cells verified the GFP data (Figure 2C). The novel bromodomain inhibitor Neo-2734 was not present in the compound library, but because of its reported potent dual activity, specifically inhibiting p300/CBP and BRD4 proteins, the drug was included in our studies [30]. Increased production of functional virus over time (24, 48 and 72 h) in combination with all three inhibitors and Ad5wt, Ad∆∆, and Ad-3∆-A20T in both Suit-2 and Panc04.03 cells confirmed the FACS and qPCR data (Appendix A). As expected, expression of the early viral E1A protein, essential for driving virus replication, was expressed at higher levels in the presence of the inhibitors (Appendix A).

### 2.4. Virus-Induced Cancer Cell Killing Is Enhanced by the OTX-015, iBet-762, and Neo-2734 Inhibitors

To assess if the increased viral production was paralleled by enhanced cell killing, the viability of Suit-2 cells was determined in the presence of drugs. All three inhibitors increased virus-mediated killing by decreasing Ad5wt EC_50_ values 9–13-fold (Figure 2D). Increased cell killing was also seen in the Panc04.03 cells and the PDAC-associated PS-1 stellate cells infected with Ad∆∆ and Ad-3∆-A20T (Figure 2E). Enhanced cell killing was also detected in Suit-2 and Panc04.03 cells infected with both oncolytic mutants in the presence of the inhibitors (Appendix A). These data showed increased adenovirus-mediated cell killing in response to bromodomain inhibitors in PDAC cells with both Ad5wt and OAds with E1ACR2 and E1B19K deletions.

### 2.5. Expression of c-Myc Is Greatly Reduced in Cells Treated with the OTX-015, iBet-762, or Neo-2734 Inhibitors

To explore whether changes in cell cycle progression contributed to the increased viral replication and cell killing, Suit-2 cells were treated with the inhibitors for 18 h after aphidicolin synchronization (Appendix A). Small increases in the G2/M-phase with corresponding decreases in the S-phase indicated that the inhibitors delayed entry into the cell cycle attenuating cell proliferation and redirecting the cellular machinery towards virus production. In support of the cell cycle findings, the cell proliferation marker Ki67 was profoundly reduced over time (24–72 h) in the presence of the inhibitors (Figure 3A). One of the most important proteins that is regulated by the BRD4-associated transcriptional machinery is c-Myc [32,35,36]. The expression of c-Myc in Suit-2 cells was greatly reduced by all three inhibitors when administered alone (Figure 3B; left blot). Virus infection resulted in increased nuclear c-Myc levels while combination treatments reduced c-Myc levels to those seen with inhibitors alone (Figure 3B; right blot). The expression of the early viral gene E1A was greatly increased by all three inhibitors both in the cytoplasm and nuclear cell compartments (right panels). Taken together, the inhibitors promote cancer cell death through increased virus replication in response to increased expression of the E1A gene. The observed decreased cell proliferation was likely the result of the inhibitors’ effect on c-Myc protein expression.

### 2.6. The Ad∆∆ Mutant in Combination with Neo-2734 Improved on In Vivo Efficacy in Suit-2 Xenograft Models

We previously reported that E1ACR2- and E1B19K-deleted mutants (Ad∆∆, Ad-3∆-A20T) are highly efficacious in PDAC xenografts [14,17]. To determine efficacy in vivo for the combination with BRD4 inhibitors, mice with Suit-2 xenografts (1 × 10^6^ cells/flank) were treated with Ad∆∆ ± Neo-2734. In these studies, we selected Neo-2734 as the representative inhibitor since similar activities and efficacy in vitro were observed in combination with all three inhibitors. A high dose of Ad∆∆ (1 × 10^10^ vp, i.t.) eliminated tumours in 90–100% of animals both alone and in combination with Neo-2734. Therefore, a lower dose of Ad∆∆ (5 × 10^8^ vp, i.t.) was administered to enable measurement of enhanced efficacy in combination with a suboptimal dose of Neo-2734 (Figure 3C; left and middle panels). In the combination treated group, 50% of animals had tumours that measured <500 mm^3^ after 52 days, while only 20% of animals in the Ad∆∆ single agent group had tumours <500 mm^3^. In the control and Neo-2734-treated groups, all animals had tumours that were >500 mm^3^ at earlier timepoints (<35 days). Importantly, all animals in the combination-treated groups were alive on day 52, had not progressed to endpoint (1200 mm^3^), were healthy, and showed significantly reduced tumour volumes compared to single agent and control-treated animals (Figure 3C; right panel). Administration of Ad∆∆ alone resulted in 60% live animals on day 52, while Neo-2734 alone and untreated animals had only 20% live animals (Figure 3C; right panel). All animals in the combination treated group responded to treatment, with three animals showing complete responses and two animals with reduced tumour growth rates up to 52 days post virus administration. Treatment with Neo-2734 alone at this low dose did not result in tumour growth inhibition (5/5 progressed) and was not different from treatment with PBS alone. The study was terminated at day 52 after virus treatment in accordance with HO regulations.

### 2.7. All Three Bromodomain Inhibitors Promote Virus Spread and Cell Killing in 3D Co-Cultures of Suit-2 Cells and PS-1 Cancer-Associated Stromal Cells

The dense PDAC tumour microenvironment (TME) is formed by a network of cancer-associated fibroblasts (CAF) and stellate cells that prevents access by therapeutic agents and immune cells [37,38]. To this end, Suit-2 cells (red) were co-cultured with tumour-associated fibroblasts (PS-1, green) in 3D-organoid models that more realistically mimic the TME in situ (Figure 4A). Organoids were infected with Ad-3∆-A20T ± each inhibitor and growth was monitored over time (Figure 4A; right). All three drugs synergized with the virus and reduced organoid growth up to 10 days after treatment. The decreased growth was caused by increased cell killing as determined by greatly reduced ATP levels in combination with the inhibitors compared to Ad-3∆-A20T infection alone (Figure 4A; right lower panel). PS-1 cells are poorly infectible and support only low levels of adenovirus replication [14]; however, the inhibitors sensitized the cells to the Ad-3Δ-A20T mutant 4–5-fold (Figure 2E). Importantly, high levels of E1A expression (blue) in the organoids were detected by confocal microscopy demonstrating that each of the three drugs increased E1A protein expression in both cancer cells and associated fibroblasts (Figure 4B). Viral genome amplification in each cell type in the organoids was examined after pre-infection with Ad-3∆-A20T of PS-1 cells or both PS-1 and Suit-2 cells (Figure 4C). The levels of replication were significantly higher when both cell types were co-infected, although the virus also spread throughout the cultures when only PS-1 cells were pre-infected (Figure 4C). Production of active viral progeny was verified under the same conditions within the organoids and was greatly increased by all three drugs (Figure 4D). As expected, the increased replication in the organoid cultures was paralleled by higher levels of E1A mRNA in the presence of the inhibitors (Figure 4E). In contrast, expression of the cancer cell migration-associated avβ6-integrin and the growth-promoting c-Myc genes was reduced in the presence of the drugs. These results indicate that BRD4 inhibitors promote adenovirus cell killing and replication not only in tumour cells but also in tumour-associated fibroblasts that are in close contact with cancer cells and suggest that the TME can be modified by these combinations to favor access of drugs, viruses, and immune cell infiltration.

### 2.8. Neo-2734 Enhanced Adenoviral Gene Expression and Replication In Vivo

To further explore whether the BRD4 inhibitors promote viral replication in vivo, a subcutaneous xenograft model of mCherry labelled Suit-2 cells was established. Replication in tumour and non-tumour tissues was monitored after intravenous administration of Ad5wtGFP (1 × 10^10^ vp/mouse ×1) and either PBS or Neo-2734 (100 µg/mouse, daily) intraperitoneally. Tumours, spleen, and liver were harvested on days 3 and 7 and processed for flow cytometry analysis (Appendix A). Ad5wtGFP in mCherry-positive tumour cells (day 3) was 5-fold higher than in mCherry-negative non-tumour cells, and in combination with Neo-2734, it was significantly increased to 18-fold higher levels compared to non-tumour cells (Figure 5A; upper panel). After 7d, Ad5wtGFP-positive cells were fewer compared to day 3 in animals treated with virus alone and showed less difference between tumour and non-tumour cells. Neo-2734 did not significantly increase virus-positive cell numbers at day 7. In contrast, viral gene expression and replication within the cells (MFI; GFP intensity/cell) was greatly increased on day 7 in tumour cells compared to day 3 in Neo-2734 treated animals; 5-fold in tumour cells and 2-fold in non-tumour cells (Figure 5A; lower panel). Increases in viral genome copies (qPCR) paralleled the GFP expression in tumour cells with up to 7-fold higher levels on day 7 in Neo-2734-treated animals (Figure 5B).

Inhibitor treatment greatly increased both E1A mRNA and protein expression in the tumour cells (Figure 5C and Appendix A). Importantly, E1A gene expression was greatly increased both at 3 days and 7 days post administration in the presence of Neo-2734, implicating higher E1A levels as the main cause of the improved viral replication and tumour elimination. Changes in mRNA expression levels of cellular growth-related genes and inflammatory effector genes were also examined (Figure 5C). Transcripts for c-Myc, Myb, CDK4, TNFα, IFNα, and IFNγ were upregulated during early infection with the virus (3d post-infection) alone but not at 7 days post-infection. This upregulation was prevented in the presence of Neo-2734 and expression continued to be low through day 7. Interestingly, expression of TGFβ was not affected by virus infection alone at any timepoint or in the presence of Neo-2734 3d post-infection, but, after 7 days, TGFβ levels were significantly increased in infected cells treated with the inhibitor. These findings demonstrate that administration of the inhibitors in vivo facilitates early viral gene expression and promotes viral spreading in tumour tissue both through direct effects on the viral transcription machinery and also indirectly through epigenetic regulation of cellular growth and immune-related factors that play roles for efficient viral propagation.

### 2.9. OTX-015, iBet-762, and Neo-2734 Inhibitors Affect Both Viral Gene Expression and Regulation of Cellular Growth and Immune Factors

To validate the gene expression changes from our in vivo studies, we carried out an extensive RNA sequencing analysis. Suit-2 cells were treated with Ad5wtGFP, iBet-762, OTX-015, or Neo-2734 alone and in combination. Virus infection alone fundamentally altered gene expression with significant changes compared to uninfected cells (Figure 6A; AdwtGFP DMSO vs. no virus DMSO). Each drug changed the gene expression to a lesser extent compared to untreated and virus-infected cells (Figure 6A, right panels and left panel AdwtGFP DMSO). Furthermore, no major differences were detected when comparing the three drugs while the gene expression patterns comparing virus-infected alone and inhibitor treatment alone were entirely different. Importantly, in combination-treated cells (virus-infection + inhibitor) the virus effects were dominant (Figure 6A, heatmap), as could be expected from the variance analysis (Appendix A). Virus infection resulted in 2731 upregulated and 3430 downregulated genes while in combination with iBet-762, and additional genes were altered (3611 upregulated and 4438 downregulated) (Figure 6B). In the combination-treated cells, 1228 upregulated and 909 downregulated genes were differentially expressed compared to single agent treatments. Populations of genes suggesting synergistic effects were identified with 344 and 741 up- and downregulated, respectively. The combinations of the virus with OTX-015 and Neo-2734 resulted in similar numbers of genes being altered (Appendix A). Based on our in vivo data we selected proliferative and inflammatory pathways for further hallmark analysis to gain more insights into global alterations in gene expression (Figure 6C–E). For example, factors representative of proliferation such as KRAS, Myc, and E2F were downregulated by iBet-762, counteracting the virus-induced expression. Similar trends were seen in cells treated with AdwtGFP and OTX-015 or Neo-2734 (Appendix A).

The RNA sequencing findings were validated by RT-qPCR in separate experiments under the same conditions in Suit-2 cells (representative genes; Appendix A). Upregulation of the viral DNA binding protein (DBP) and E1A expression were verified by increased mRNA levels. In contrast, Akt3, Bcl-2, FOSL-1, αvβ6-integrin, and TLR3 and 4 were downregulated. As expected, the expression of c-Myc, the main target for BRD4 proteins was significantly suppressed on both the mRNA and protein levels in the presence of inhibitors (Figure 3B and Figure 4E; Appendix A). Taken together, these findings indicate that the BRD4 inhibitors promote viral gene expression and replication while attenuating cellular factors that typically would hinder viral propagation such as Akt, Bcl-2, and the toll-like receptors. Furthermore, in vivo data demonstrated a downregulation of host immune defence pathways including IFNγ, IFNα, and TNFα, and cell cycle related factors including CDK4, Myb, and c-Myc (Figure 5C). These results suggest that the effects of the inhibitors on both adenovirus replication and cellular gene expression contribute to the strong synergistic cancer cell killing when combined both in vitro and in vivo.

## 3. Discussion

In this study, we demonstrate that the potency of OAds was markedly improved in combination with the bromodomain inhibitors iBet-762, OTX-015, and Neo-2734 in preclinical models of PDAC. The increased efficacy was due to higher levels of E1A expression followed by amplified viral replication and cell killing in all in vitro and in vivo models. These findings indicate that OAd combinations with inhibitors of bromodomain proteins may be clinically more beneficial than combinations with standard of care cytotoxic drugs that frequently attenuate viral efficacy. The mode of action for cytotoxic drugs such as gemcitabine, used to treat PDAC patients, is to induce cancer cell death through DNA damage. However, the drugs also cause damage to the adenoviral DNA, resulting in significantly reduced viral replication, spread, and efficacy [13,14]. Infection with the replication-selective Ad∆∆ and Ad-3∆-A20T alone induced cell killing and tumour regression in PDAC models, while treatment with the BRD4 inhibitors alone had no effect at the doses used in this study. Importantly, viral replication and cell killing were significantly increased in both PDAC and stellate cells treated with the BRD4 inhibitors compared to non-treated cells both in 2D and 3D cultures in vitro and in vivo xenografts.

Screening of the small molecule epigenetics compound library resulted in the novel identification of bromodomain proteins as regulators of adenoviral functions. The BRD4 inhibitors iBet-762, OTX-015, and Neo-2734 proved to significantly enhance replication with no or minor effects on viral uptake. At later time points after infection (48–72 h), increased uptake was observed as a consequence of the higher levels of viral replication after the first round of infection, supporting continued replication and spread throughout both 2D and 3D cultures. Importantly, cell viability was decreased with all viruses in combination with BRD4 inhibitors, not only in PDAC Suit-2 and Panc04.03 cells but also in the transformed stellate cell line PS-1 (normally insensitive to adenovirus).

Bromodomain proteins primarily bind to acetylated histone tails and function as transcriptional coactivators with key roles in tumourigenesis and cell growth. The inhibitors block the binding of BRD4 proteins to acetylated histones, resulting in the downregulation of several oncogenes in PDAC cells and direct inhibition of tumour growth [40,41,42,43,44]. One of the main target genes of BRD4 proteins is c-Myc, which is a well-known oncogene in multiple tumourigenesis pathways [35,36]. Downstream target genes regulated by c-Myc are primarily involved in proliferation, differentiation, cell cycle progression, metabolism, apoptosis, and angiogenesis. We found that expression of c-Myc was significantly suppressed on both the mRNA and protein levels in samples from the in vivo study in response to the bromodomain inhibitor Neo-2734. Other tumour growth-related genes, such as Myb, KRAS, Akt-3, and CDK4 were also downregulated by the three BRD4 inhibitors in PDAC cells. Adenovirus infection leads to global alterations of the host cell transcription machinery to favour the viral life cycle, ultimately lysing the infected cell and spreading to surrounding cells and tissues. Inevitably, the infected host cell responds to Ad-infection by activation of inflammatory-related pathways including innate and adaptive immune responses. We found that mediators of the interferon response including IFNγ and IFNα, and factors such as TNFα and TGFβ, were greatly downregulated in combination with the inhibitors. In contrast, virus infection alone induced the expression of these factors, ultimately attenuating viral propagation.

Cell viability data showed decreased proliferation in the presence of the inhibitors accompanied by decreases in S-phase and increases in G1 (Neo-2734) or G2/M cell populations. OAd replication requires the recruitment of host transcription factors in tumour cells; in BRD4 inhibitor-treated cells, the absence of specific acetylated cellular histones may shift the location of transcription factors to favor the viral life cycle. Our data support this hypothesis by demonstrating strong viral early gene expression in response to the drugs, which boosts OAd replication, finally resulting in potent tumour cell killing by effective oncolytic virion production.

Interestingly, the increased viral replication and cell killing were independent of whether cells were infected with Ad5wt, Ad∆∆, or Ad-3∆-A20T. This finding suggests that the inhibitors facilitated early viral gene transcription/expression and functions that are independent of the domains deleted in the OAds; E1ACR2 and E1B19K. The expression of E1A was greatly increased in the presence of the inhibitors in combination with all three viruses and was not caused by increased viral uptake but rather increased transcription. Our findings are in agreement with a previous report demonstrating that another BRD4 inhibitor, JQ1, enhanced Adenovirus type 2 (Ad2) gene expression and gene delivery in the human A549 cells [45]. The authors found that the enhancement was due to transcription-mediated events without affecting the Coxsackievirus and Adenovirus Receptor (CAR) expression or viral uptake. The expression of E1ACR2 is an absolute requirement for viral replication in normal cells and redirects the cellular DNA and protein synthesis machinery to support viral gene expression and virion production [17]. However, additional E1A sequences are essential for productive infections in both healthy and cancer cells [24,26]. For example, the E1ACR1 and N-terminal protein domains play important roles in the recruitment and binding of numerous cellular factors that include histone acetyltransferases such as p300/CBP, p400, and pCAF, and transcription-related proteins, for example, TBP [26,27,46]. Our data indicate that the BRD4 inhibitors altered the association of essential transcription factors with the cellular chromatin towards viral DNA by promoting interactions with the E1ACR1 or other N-terminal regions which were present in all tested viruses.

To our knowledge, BRD4 proteins have not been demonstrated to bind E1A directly while bromodomain binding partners, mainly the transcription factor c-Myc, are linked to successful viral propagation. One example is the E1A binding to p400 that stabilizes c-Myc and promotes localisation to chromatin, thereby activating target genes [47]. By inhibiting the BRD4 c-Myc binding, the E1A p400 c-Myc association may be promoted through the activation of a different set of genes than with BRD4 c-Myc. Another example is E1A binding to p300/CBP, which relocates the complex from active cellular promoters to promoters essential for viral replication such as for regulation of cell cycle entry [27]. Although the Neo-2734 inhibitor blocks both p300/CBP and BRD4 activities, we did not detect a significant increase in viral replication compared to the BRD4-only inhibitors (iBet-762, OTX-015) [30]. However, it is important to consider that during infection the E1A binding to p300/CBP varies over time and further studies should explore additional benefits of Neo-2734 administered at different time points during the viral life cycle. Our data demonstrate that the inhibition of the cellular transcription of specific genes regulated by bromodomain proteins on an epigenetic level, such as c-Myc, promotes viral replication both through effects on cellular genes and through increases in early viral gene expression.

The findings reveal that the effects of the BRD4 inhibitors on OAd replication, oncogene expression, and inhibition of inflammatory pathways contribute to the increased cancer cell killing and amplified viral replication in PDAC models. Furthermore, we demonstrated that the combination of OAd and BRD4 inhibitors increased cell lysis and viral replication also in the virus-insensitive stellate cells (PS-1). Taken together, these observations suggest that the combination treatment may effectively lyse PDAC and associated stromal cells and break through the desmoplastic PDAC TME in situ to pave the way for host anti-tumour immune cells and therapeutics. Complete tumour elimination, especially in patients with late-stage metastatic lesions, requires the recruitment of an antigen-dependent immune response through collaboration between CD4+/CD8+ T cells and B cells, as well as the production of cytokines. Previous reports demonstrated that inactivation of c-Myc induced disassembly of advanced PDAC tumours and stroma, and downregulated PD-L1 expression in the TME, leading to the restoration of antitumour responses through influx of immune cells and efflux of immune suppressive cells [48,49]. Tumour cell death triggered by drugs and enhanced oncolytic virus replication will release abundant neoantigens within the tumour and form an immune-favourable TME. Our data encourage further development for future clinical applications, especially in combination with immune therapies to generate a specific and lasting adaptive immune response to ultimately improve outcomes for PDAC patients.

## 4. Materials and Methods

### 4.1. Cell Lines and Adenoviruses

The human HEK293 and subclone JH293 (embryonic kidney cells), A549 (alveolar basal adenocarcinoma), Panc04.03, and Suit-2 (pancreatic ductal adenocarcinoma; PDAC) cell lines (ATCC, Manassas, VA, USA) were cultured in DMEM (Dulbecco’s Modified Eagle’s Medium; Invitrogen, Paisley, UK) and PS-1 cells (hTERT-immortalised pancreatic stellate cells; Prof. H. Kocher, BCI, QMUL, London, UK) were cultured in DMEM/F12 medium supplemented with 10% foetal bovine serum (FBS), 100 µg/mL penicillin, 50 µg/mL streptomycin, and 2 mM L-glutamine. Cells were derived from STR-verified stocks and verified to be free of mycoplasma.

All viruses were derived from human adenovirus subtype C serotype 5 (HAdVC5), including Ad5wtGFP (wild type Ad5 expressing GFP under control of the E3 promoter), Ad∆∆ (deletion of E1ACR2 and the E1B19K gene), and Ad-3∆-A20T (Ad∆∆ with E3gp19K deleted and the FMDV peptide expressed from the fibre knob) [14,17,20]. The viruses were propagated in A549 or JH293 cells, purified by discontinuous CsCl_2_-gradient centrifugation and dialysed overnight as previously described [17]. Virus yield was determined by DNA quantification (Qubit, ThermoFisher, Waltham, MA, USA) after protein digestion and converted to viral particles/mL (vp/mL). Viral activity was determined by tissue culture infectious dose (TCID_50_) and expressed as plaque-forming units/mL (pfu/mL). Typical ratios were 2–50 vp/pfu.

### 4.2. High Throughput Screening (HTS) Assay for Adenoviral Replication

Each compound in the library, a unique collection of 181 molecules enriched in epigenetics compounds, (Cat L1900, Selleckchem, Houston, TX, USA) was resuspended in DMSO as 10 mM and 100 µM stocks and barcoded. A Hamilton Microlab Star liquid handling platform (SPT Labtech, Melbourn, UK) was used for further dilutions to create 384 well-plates of stocks in DMSO. Suit-2 cells (800 cells/well, in 50 µL) were seeded in 384-well flat bottom plates in 10% FBS-DMEM and cultured for 40 h when the media were replaced by 45 µL 2% FBS-DMEM before the addition of the compounds. The robotic protocol design, scheduling, and execution were carried out using the Echo Plate Reformat Application and Echo Liquid Handler instrument (Labcyte Ltd., Sheffield, UK). Drugs were dispensed at final concentrations of 10 nM (5 nL of 100 µM stock), 100 nM (50 nL of 100 µM stock), 500 nM (2.5 nL of 10 mM stock), 1 µM (5 nL of 10 mM stock), and 5 µM (25 nL of 10 mM stock) into individual wells in triplicate plates for each dose. Ad5wtGFP 500 particles per cell (ppc) in 5 µL were added into each well manually. DMSO (0.1%) was added to untreated control cells, positive controls were infected with the virus pre-treated with 10 mM L-lysine and negative controls with 100 nM Gemcitabine. Assay plates were placed in an incubator at 37 °C for 24 h and the Hoechst 33,342 dye (10 μg/mL) was added and incubated for an additional 20 min. The plates were imaged by the Celigo Imaging cytometer (Nexcelom Bioscience, Lawrence, MA, USA); cellular DNA (Hoechst dye) and GFP (AdwtGFP) were detected by excitation at 350 nm/emission at 455 nm and excitation at 490 nm/emission at 525 nm, respectively. GFP expression was quantified as an indirect measure for early viral gene expression and replication per viable cell (Hoechst dye) using the Celigo system (nexelom.com). The quantitative numerical output was % GFP+ cells, total live cell numbers (Hoechst), mean GFP density/well, and integrated mean GFP density/well. All data from triplicate plates were averaged and calculated. Significance was determined by the Z factor according to the following: Z-score = (mean for individual drug − mean for all drugs treated and untreated wells)/STD from all drug-treated and -untreated wells. Samples with Z-scores greater than 2 were selected for a second round of screening analysis by qPCR.

### 4.3. DNA Extraction and qPCR Validation

Cells in the 384-well plates were lysed (30 µL/well; 10 mM Tris-HCl pH 7.5, 10 mM EDTA pH 8.0, 10 mM NaCl, 0.5% sarcosyl, 1 mg/mL proteinase K), incubated at 55 °C for 2 h. Lysates were transferred to 96-well plates and DNA precipitated, resuspended, and quantified by qPCR using SYBR Green and E2A specific primers: forward (5′-GGATACAGCGCCTGCATAAAAG-3′) and reverse (5′-CCAATCAGTTTTCCGGCAAGT-3′). The human GAPDH gene was used as a reference in all reactions with the following primers; forward (5′-TGGGCTACACTGAGCACCAG-3′) and reverse (5′-GGGTGTCGCTGTTGAAGTCA-3′). Amplified DNA was analysed by the comparative Ct method (∆∆Ct) and expressed as viral particles per cell (vp/cell). To validate the drug effects in separate experiments, Suit-2 cells (1 × 10^4^ cells/96-well) were treated with selected drugs at doses identified in the HTS screen and primary qPCR, 6 wells/dose. Cells were infected with AdwtGFP (20 ppc) and incubated at 37 °C for 24 h. DNA was extracted and quantified by qPCR (as above).

### 4.4. Viral Replication Assay by Tissue Culture Infections Dose (TCID_50_)

Suit-2 cells (2 × 10^5^/6-well) were infected with Ad5wtGFP or Ad5wt (50 ppc) for 2 h, treated with the inhibitors and harvested after 4, 24, 48, and 72 h. Cells and media were freeze–thawed prior to assay; 3 cycles at N_2(l)_ and 37 °C and diluted prior to serial dilutions on the detector JH293 cells. Cytopathic effects (CPE) were determined after 10–14 days and virus titres were calculated as previously described [17].

### 4.5. Cell Viability Assay

Cells (10,000 cells/96-well) were infected with the virus or treated with drugs or a combination of virus and drugs in 2% FBS/1% P/S DMEM. Cell viability was determined after 3–6 days by the MTS assay (3-(4,5-dimethylthiazol-2-yl)-5-(3-carboxymethoxyphenyl)-2-(4-sulfophenyl)-2H-tetrazolium assay; Promega, Southampton, UK) and live cells were detected by a microplate reader (SPECTROstar; BMG Labtech, Ortenberg, Germany). Viral and drug dose–response curves were generated to determine the concentration killing 50% of cells (EC_50_) using untreated and drug-treated cells as controls. Each data point was generated from triplicate samples and experiments repeated at least three times [17].

### 4.6. Immunoblotting of Nuclear and Cytosolic Proteins

Cell pellets were re-suspended in 500 µL hypotonic buffer (20 mM Tris-HCl, pH 7.4, 10 mM NaCl, 3 mM MgCl_2_) and incubated on ice for 15 min. NP40 (0.5%) was added and suspension vortexed for 10 s, nuclei were pelleted at 3000 rpm at 4 °C for 10 min, and cytosolic fractions collected. Pellets were lysed (1 mM Tris, pH 7.4, 10 mM NaCl, 1 mM Na_3_VO_4_, 1 mM EDTA, 1 mM EGTA, 1 mM NaF, 20 mM Na_4_P_2_O_7_, 0.5% deoxycholate, 0.1% SDS, 1% Triton X-100, 10% glycerol, 1 mM PMSF, protease inhibitor cocktail, and PhosSTOP phosphatase inhibitor (Roche Diagnostics, Rotkreuz, Switzerland)) on ice for 30 min and centrifuged at 14,000× *g* at 4 °C for 10 min. Protein concentrations were determined by the BCA reagent (Pierce, Vallejo, CA, USA) and proteins (10–20 µg/lane) were resolved on SDS Polyacrylamide Gel Electrophoresis (12%; SDS-PAGE) and transferred to Immobilon-P PVDF membranes (Merck, Rahway, NJ, USA). Membranes were incubated with primary antibodies at 4 °C overnight, washed, and incubated with secondary HRP-conjugated antibodies (Dako, Copenhagen, Denmark) for 1 h at 24 °C, followed by detection using the enhanced chemiluminescence substrate ECL (PerkinElmer, Shelton, CT, USA) and images were captured by the Amersham 600 imager (GE Healthcare, Bio-Sciences AB, Uppsala, Sweden).

### 4.7. Isolation of RNA and Analysis of mRNA Expression

Total RNA was extracted from treated cells by the Trizol reagent (Invitrogen) and concentrations were determined by NanoDrop Spectrophotometer (ThermoFisher). Total RNA (200–1000 ng) was used for cDNA synthesis by SuperScript III reverse transcriptase following the manufacturer’s protocol (Invitrogen). The synthesised cDNA was amplified and quantified according to the procedure described above for validation and mRNA expression normalized to β-actin; forward (5′-CTTCGCGGGCGACGAT-3′), reverse (5′-CCACATAGGAATCCTTCTGACC-3′).

### 4.8. Organoid Co-Cultures

The PS-1-GFP and Suit-2-mCherry cells were established by cell sorting after infection with replication-deficient retrovirus carrying the orf for the respective fluorescence proteins. Both cell lines were infected with Ad∆∆ (2 ppc) in serum-free DMEM for 2 h, followed by replacement of infection media with 10% FBS/DMEM. After an additional 2 h, cells were trypsinised and 9000 cells at ratio of 2:1 (PS-1-GFP/Suit-2-mCherry) were plated into each well in ultra-low attachment 96-well plates in 50 µL 10% FBS/DMEM. The plates were centrifuged and placed in an incubator overnight. The following day, 50 µL Matrigel (Merck) at 1.8 mg/mL in 10% FBS/DMEM was overlayed onto organoids and centrifuged to maintain organoid position and placed in an incubator to solidify the Matrigel. Drugs were added in 100 µL 10% FBS/DMEM on top of the gels and plates were centrifuged. Organoid growth was monitored daily using Incucyte S3 for the indicated time in each study.

Organoids were harvested in 50 µL medium after 5–6 days of drug treatment, ATP extraction buffer (50 µL; Abcam, Cambridge, UK) was added, and incubated at 24 °C for 20 min. The ATP-extracted cell solution (2 µL/well) from each organoid was added to ddH_2_O (48 µL) in a 96-well luciferase assay plate, followed by the addition of CellTiter-Glo luminescent cell visibility assay reagent (50 µL; Promega). The reaction mixture was incubated at 21 °C with shaking for 20min and the luciferase signal was quantified using a FLUOstar Omega microplate reader (BMG Labtech, Aylesbury, UK).

Organoids were fixed in 4% PFA at 24 °C for 1 h, washed ×3 with cold PBS, permeabilized, and non-specific binding blocked with 5% normal donkey serum in 0.5% Triton X-100/PBS at 4 °C overnight. Proteins were detected by mouse anti-Ad5 E1A (1:5 00; Invitrogen, Waltham, MA, USA) followed by donkey anti-mouse Alexa 405 conjugated antibody (1:1000; Dako, Copenhagen, Denmark). Organoids were washed 3× PBS, mounted, and images recorded using a spinning disc confocal microscope (Nikon CSU-W1 SoRa, Tokyo, Japan) and analysed by Imaris imaging data analysis software (version 9.7).

### 4.9. In Vivo Tumour Growth and Bio-Distribution

Suit-2 or Suit-2-mCherry cells (1 × 10^6^ cells 50 µL in 50 µL Matrigel) were inoculated subcutaneously in one flank of female BALB/c *nu*/*nu* mice (Charles River, Harlow, UK) when animals were 6 to 8 weeks, as previously described [14,17]. Treatments were initiated when the tumours reached 100 ± 20 mm^3^. Suboptimal doses of Ad∆∆, 5 × 10^8^–1 × 10^10^ vp in 50 µL/injection, were administered intratumourally (IT) 3 times on day 1, 3, and 5 ± suboptimal doses of Neo-2734 in 100 µL (1 µg/µL) administered intraperitoneally 5 days a week for 5 weeks. Animals were sacrificed when tumour volume reached 1.25 cm^3^ (length × width^2^ × π/6) or if there was the development of clinical signs (according to animal welfare regulations; UK Home Office). Tumour growth rates were calculated and analysed by one-way Anova, and survival analysis was performed according to the method of Kaplan–Meier (log-rank test for statistical significance), n = 5/group, and two studies. For bio-distribution studies, the virus was administered via the tail vein once, 3 weeks after tumour cell inoculation; Ad5wtGFP 1 × 10^10^ vp/100 µL ± 100 µL Neo-2734 (1 µg/µL) or PBS intraperitoneally (IP) daily and tissue collected at 3 days and 7 days post virus administration. Tumours and non-tumour (spleen and liver) samples were processed for FACS analysis and DNA and mRNA extraction was performed for further analysis, n = 3/group.

### 4.10. RNA Sequencing

Suit-2 cells were treated with iBet-762 (2 µM), OTX-015 (1 µM), Neo-2734 (0.5 µM), or DMSO (ctrl) for 12 h alone or infected with Ad5wtGFP for 2 h alone, or inhibitors were added to the infected cells after 2 h and cultured for another 12 h. Ad5wtGFP-infected cells were harvested and GFP+ cells were collected by FACS sorting. Total RNA was extracted using the Trizol reagent (Invitrogen), quantified, and RNA integrity determined by Tapestation (Agilent, Santa Clara, CA, USA). Samples with an RNA Integrity Number (RIN) >8.0 were used for the RNA sequencing library preparation and mRNA was enriched by Poly(A) mRNA magnetic beads (New England BioLabs, Ipswich, MA, USA). The NEBnext Ultra II RNA Library Prep Kit for Illumina (E7770, San Diego, CA, USA) was used to generate RNA sequencing libraries. Libraries were sequenced for 37 bp paired end by NextSeq 500 high output Run, with an average of 30 million reads per sample.

### 4.11. RNA-Seq Data Analysis

Base calls, demultiplexing, and adapter trimming were performed with Illumina reagents (Illumina, Cambridge, UK). A custom reference genome consisting of the GRCh38 build of the human reference genome obtained from GENCODE (Ensembl 104) and the sequence of the Ad5wtGFP expression construct, which is derived from the Human Adenovirus C5 (Genbank accession number: AC_000008_v1), was created. The short sequence reads were mapped to this custom reference genome using the spliced aligner Hisat2 (version 2.2.0) [50]. Intermediate processing steps and marking of optical duplicates was performed using Samtools (version 1.10) [51]. Reads mapped to each gene were quantified using htseq-count (version 0.13.5) [52]. Reads marked as optical duplicates were quantified separately and subtracted from the total counts for each gene using R (version 4.0.5). Batch correction was performed with ComBat-seq [53] from the R package svn (version 3.38.0) [54]. To identify genes differentially expressed between groups, we used the R/Bioconductor package edgeR (version 3.32.1) [55,56]. Briefly, count data were first normalized and dispersion estimated before a negative binomial model was fitted with significance assessed by a quasi-likelihood F-test [57]. Resulting *p*-values were adjusted for multiple testing using the Benjamini–Hochberg procedure. Genes with an adjusted *p*-value less than or equal to 0.05 and an absolute fold change greater than or equal to 1.5 were considered differentially expressed.

For the heatmap, a variance stabilizing transformation from the DESeq2 (version 1.30.1) and vsn (3.58.0) packages [58] was applied to the dataset and differentially expressed genes selected. Z-scores were calculated for each gene before clustering using 1—Pearson correlation as a distance metric and visualization with the ComplexHeatmap package (version 2.6.2) [59]. For functional annotation, the msigdbr package (version 7.5.1) [39] was used to obtain MSigDB gene sets and pre-ranked GSEA was performed with the clusterProfiler package (version 3.18.1) [60] using the quasi-likelihood F-statistic multiplied by the sign of the log2 fold change as the ranking metric.

## 5. Conclusions

Our findings reveal that inhibitors of bromodomain proteins (specifically BRD4) promote OAd replication and increase virus-mediated cancer cell killing both in vitro and in vivo in preclinical models of PDAC by greatly increasing early viral gene expression, especially the essential E1A protein. The inhibitors act epigenetically by reducing oncogene expression, including c-Myc, and decreasing expression of immune-related proteins such as those regulating the interferon response pathways. Furthermore, combining OAd and the bromodomain inhibitors increases cell lysis and viral replication also in the virus-insensitive stellate cells (PS-1). Taken together, these observations suggest that the combination treatment may effectively lyse PDAC and associated stromal cells and break through the desmoplastic PDAC TME in situ to pave the way for access by the host anti-tumour immune cells and therapeutics.

## Figures and Tables

**Figure 1 ijms-25-01265-f001:**
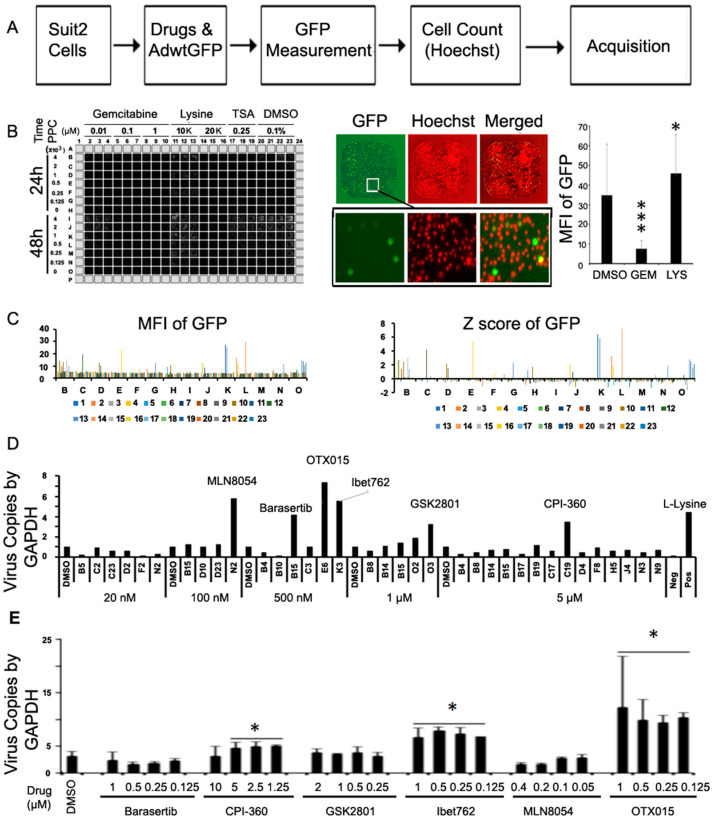
High throughput screening (HTS) of a small molecule compound library to identify molecules that increase oncolytic virus replication. (**A**) Overview of the screening of small molecule inhibitors in the PDAC Suit-2 cell line. Cells were plated in 384-well plates and incubated for 40 h (800 cells/well). The inhibitors were added at 10 nM–5 µM and Ad5wtGFP at 500 particles/cell (ppc). After 24 h exposure, viral gene expression (GFP) in live cells (Hoescht positive) was quantified. (**B**) Example of representative plate with cells treated with Gemcitabine (100 nM; inhibition of gene expression/replication) and Lysine (10 mM; increase of viral uptake/early viral gene expression) served as negative and positive controls, respectively. Whole plate view for GFP expression at increasing viral doses (125–4000 particles/cell (ppc) ((**B**) left panel). One representative well with GFP and Hoechst signals ((**B**) middle panel), upper row shows entire well and lower row the respective magnification of the indicated area). ((**B**) right panel) average MFI for GFP in individual wells treated with Gemcitabine (GEM) or Lysine (LYS), n = 3, * *p* < 0.05, *** *p* < 0.001 compared to DMSO alone. (**C**) Average GFP expression (left) in all wells from screen with 0.5 µM drugs and the corresponding Z-scores (right), data from triplicate plates. (**D**) Validation of viral genome copies in selected wells (Z-score > 2) by qPCR. (**E**) Validation of virus replication in separate experiment by quantifying viral genome amplification (qPCR) in response to the selected compounds. Suit-2 cells were treated with the identified inhibitors at increasing doses and analysed after 20 h of infection with AdwtGFP (1 pfu/cell), viral E2A gene normalised to GAPDH, n = 3, * *p* < 0.05, compared to virus infection in DMSO alone.

**Figure 2 ijms-25-01265-f002:**
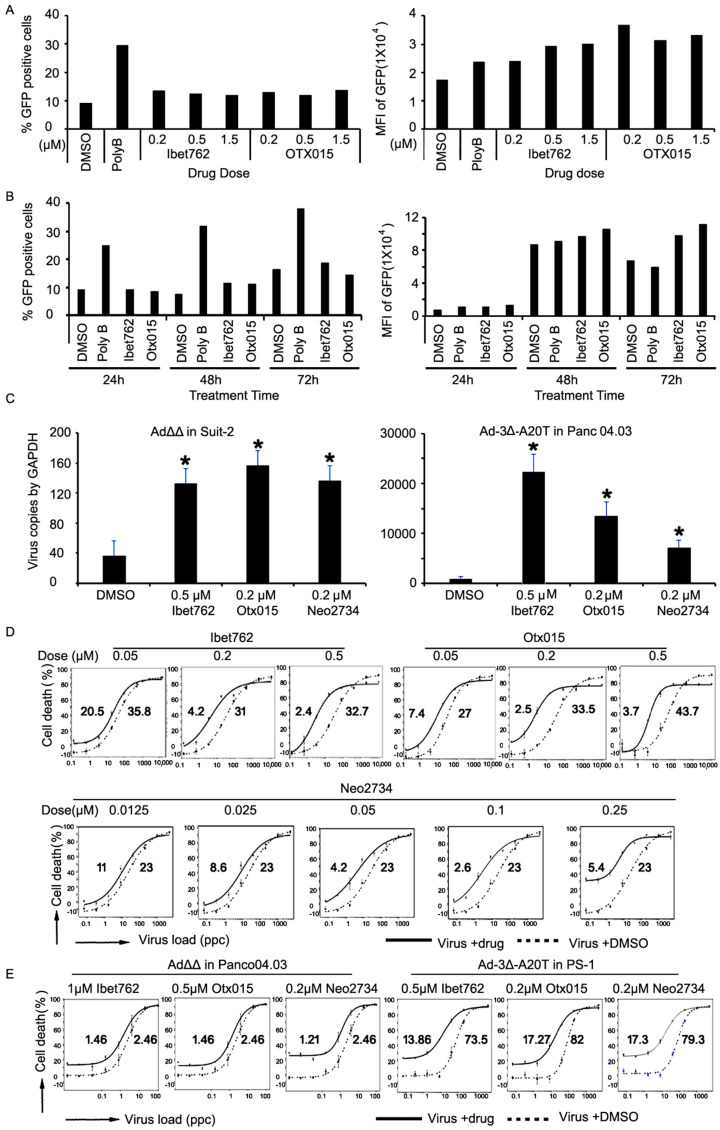
The bromodomain inhibitors iBet-762, OTX-015, and Neo-2734 promote adenovirus replication and cell killing. (**A**,**B**) Suit-2 cells were infected with AdwtGFP at 1 pfu/cell, treated with the inhibitors at the indicated doses, and analysed by flow cytometry after 24 h (**A**) and from 24–72 h (**B**). (Left panels), percentage of GFP positive cells, (Right panels), mean fluorescence GFP intensity. Polybrene at 6 µg/mL was used as positive control for adenovirus transduction. Averages from triplicate samples, one representative experiment from three studies. (**C**) Amplification of viral genomes by qPCR in the presence of the inhibitors at the optimal concentrations and infected with Ad∆∆ or Ad-3∆-A20T at 1 pfu/cell. Viral E2A DNA was quantified relative to cellular GAPDH 24 h post-infection, n = 3, * *p* < 0.01. (**D**) Suit-2 cells were infected with Ad5wt at increasing doses (0.1–10,000 ppc) ± the inhibitors at the indicated dose levels and viability was determined after 6 days. (**E**) Panc04.03 cells were infected with Ad5∆∆ and PS-1 cells with Ad-3∆-A20T at increasing doses (0.001–100 ppc) ± the inhibitors at the indicated dose levels and viability was determined after 5 days. (**D**,**E**) Data representative of three individual experiments, expressed as % live cells and EC_50_ values indicated on each graph above (virus + drug) and below (virus alone) the respective line.

**Figure 3 ijms-25-01265-f003:**
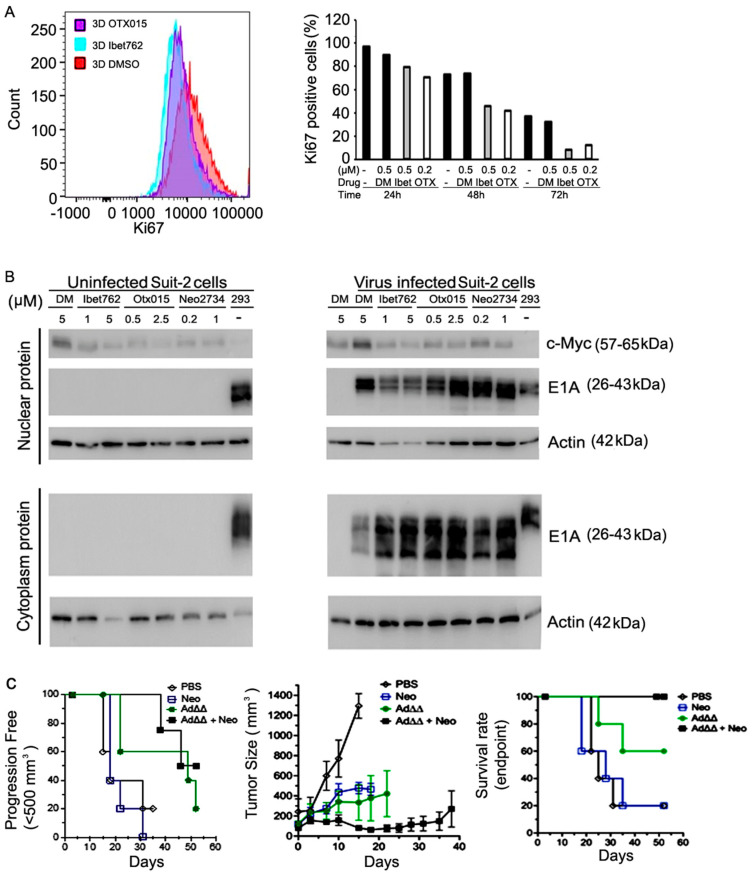
The bromodomain inhibitors attenuate c-Myc expression and promote Ad∆∆ efficacy in Suit-2 xenografts in vivo. (**A**) Suit-2 cells were treated with DMSO (ctrl), iBet-762 (0.5 µM), or OTX-015 (0.2 µM) for 24, 48, and 72 h, followed by Ki67 staining and flow cytometry. (Left panel), flow cytometry profile from one representative experiment, (right panel), quantification of Ki67 flow cytometry data over time with increasing concentrations of drugs (representative data, duplicate wells), DM = DMSO. (**B**) Non-infected and infected (Ad5wt) Suit-2 cells were treated with the iBet-762, OTX-015, or Neo-2734 inhibitors at the indicated doses for 24 h. Cells were harvested and cytoplasmic and nuclear protein fractions analysed by immunoblotting (representative blots). c-Myc also served as a nuclear marker, while the early viral gene E1A is highly expressed in the cytoplasm and rapidly enters the nuclei. (**C**) Neo-2734 promotes Ad∆∆-mediated growth inhibition of Suit-2 subcutaneous xenografts in vivo. Ad∆∆ (1 × 10^8^ vp/dose) was administered intratumourally on days 1, 3, and 5, with or without Neo-2734 (1 µg/µL) administered intraperitoneally on days 1, 2, 3, 4, and 5. (Left panel), progression free survival (tumours < 500 mm^3^), n = 5 animals/group. Middle panel; average tumour growth curves, n = 5 animals/group. Right panel; overall survival at endpoint (1200 mm^3^; HO regulations), n = 5 animals/group.

**Figure 4 ijms-25-01265-f004:**
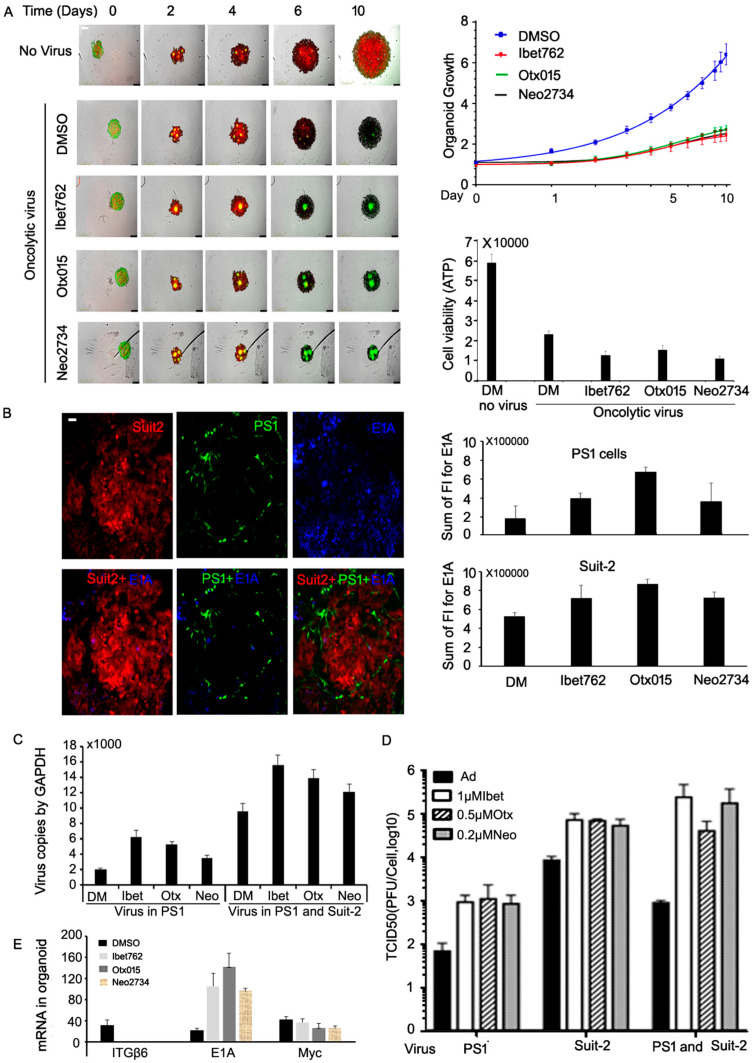
The bromodomain inhibitors promote virus spread and cell killing in 3D co-cultures of Suit-2 PDAC cells and cancer-associated stromal cells PS-1. (**A**) Growth of Suit-2 cells expressing mCherry (red) and PDAC-associated stromal cells (PS-1) expressing GFP (green) were infected with the oncolytic Ad-3∆-A20T (0.2 ppc) for 2 h ± iBet-762 or OTX-015 or Neo-2734 and embedded in Matrigel. Images were recorded daily by Incucyte up to 10 days; scale bar = 400 µm. (Right upper panel), growth curves corresponding to image data. (Right lower panel), ATP was extracted from the organoids 6 days after inhibitor treatment and analysed by luciferase assay as a marker for cell viability. (**B**) Spinning disc confocal microscopy images of organoid cultures. Organoids were harvested at 6 days after inhibitor treatment, fixed, permeabilized, and stained for viral E1A (blue); scale bar = 100 µm. Images were recorded and analysed for relative E1A protein expression (right panels) in both Suit-2 (red) and PS-1 cells (green). (**C**) Organoids were pre-infected with Ad-3∆-A20T, either in PS-1 cells only or in both PS-1 and Suit-2 cells. DNA was extracted from the organoids 6 days after inhibitor treatment and viral genomes were determined by qPCR, n = 3. (**D**) Viral replication in organoids was determined by TCID_50_. Either Suit-2 cells or PS-1 cells or both cell lines were pre-infected with Ad∆∆ (0.2 ppc) for 2 h and treated with the inhibitors for 6 days in organoid cultures. Cells and culture media were collected and analysed by TCID_50_, n = 3. (**E**) mRNA expression in organoids. Total RNA was extracted from organoids 6 days after inhibitor treatment and mRNA expression for E1A, c-Myc, and αvβ6-integrin (ITGβ6) was analysed by qPCR, n = 3.

**Figure 5 ijms-25-01265-f005:**
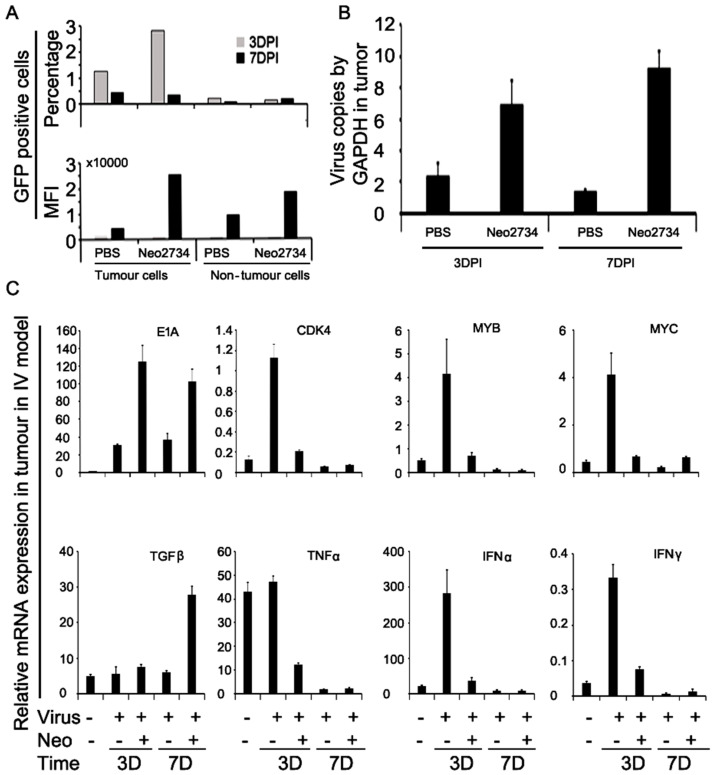
Neo-2734 increases viral replication and distribution in Suit-2 subcutaneous xenografts in vivo. Suit-2 cells (5 × 10^6^) expressing mCherry were implanted subcutaneously in Matrigel (1:1) in NOD2SCID mice. Ad5wtGFP virus (1 × 10^10^ vp/dose) was administered intravenously when tumours were 100 ± 20 mm^3^. Neo-2734 (1 µg/µL) or PBS was administered intraperitoneally. Animals were culled at 3 days and 7 days post-infection/treatment, and tumours, liver, and spleen were harvested and processed for analysis. (**A**) (Upper panel), percentage of GFP-expressing cells at 3 days and 7 days post-infection in tumour and non-tumour tissues (liver and spleen). (Lower panel), mean fluorescence intensity in GFP positive cells as an indication of viral replication/spread in tumour and non-tumour tissues. (**B**) Virus replication analysed by qPCR with E2A primers in tumour cells harvested 3 days and 7 days after treatment, n = 3/group. (**C**) Viral and cellular gene expression in tumour tissue 3 days and 7 days post-infection, mRNA analysis by RT-qPCR, n = 3/group.

**Figure 6 ijms-25-01265-f006:**
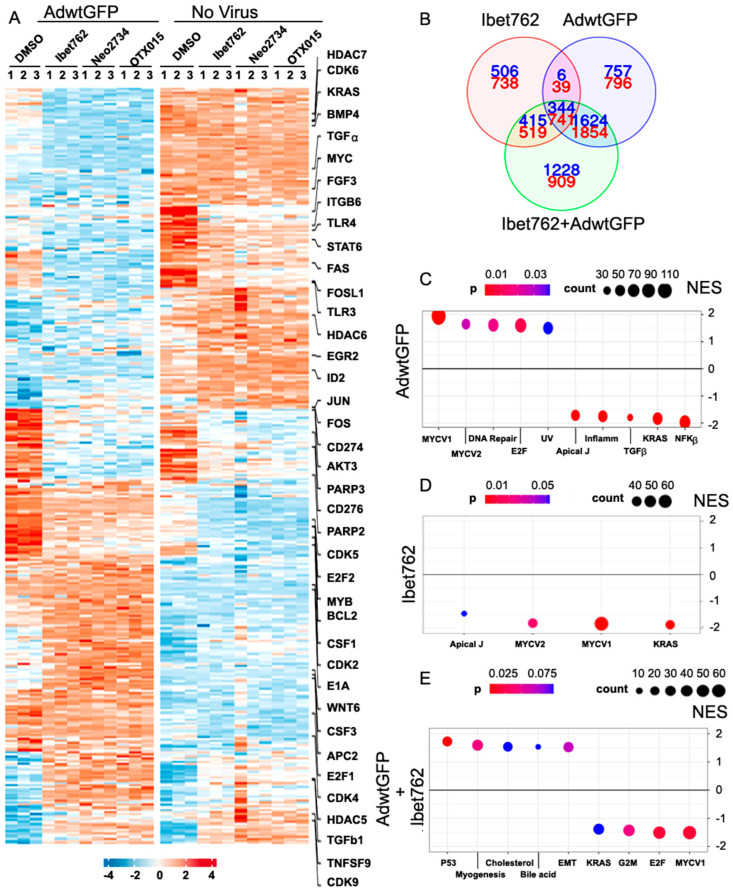
Global effects of Ad5wtGFP and bromodomain inhibitors on viral and cellular transcriptomes. (**A**) The PDAC Suit-2 cells were infected or uninfected by Ad5wtGFP for 12 h and treated with the inhibitors for another 12 h. GFP-positive cells were isolated by fluorescence activated cell sorting. Total RNA was extracted and subjected to RNA sequencing in biological triplicate samples for each treatment. The heatmap shows the global effects of the virus and inhibitors on the transcriptomes of both cancer cells and adenoviral genes. (Left panel), effects of virus alone in DMSO (left lane) and virus + inhibitors in DMSO (right lanes) compared to control treated cells (DMSO only). (Right panel), effects of each inhibitor alone in DMSO compared to control treated cells (DMSO) on global gene expression. (**B**) Venn diagram illustrating the interactions of iBet-762, virus, and the combined treatment; red and blue numbers indicate downregulated and upregulated genes, respectively. (**C**–**E**) Data analysed by GSEA using Broad Institute Hallmark gene sets [39]. Normalised enrichment scores (NES) and Benjamini–Hochberg adjusted *p*-values are shown. Positive NES indicate upregulated pathways while negative scores denote that the pathway is downregulated by treatment. AdwtGFP infection alone (**C**), iBET-762 alone (**D**), and the combined effects (**E**).

## Data Availability

The RNA-sequencing data are available from the ArrayExpress website (www.ebi.ac.uk/arrayexpress) under accession number E-MTAB-13108: https://www.ebi.ac.uk/biostudies/arrayexpress/studies/E-MTAB-13108?key=e7980c66-5fe6-4798-b564-e272d7564b18, (accessed on 1 November 2022).

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
