# Peer review of "Inhibition of Bromodomain Proteins Enhances Oncolytic HAdVC5 Replication and Efficacy in Pancreatic Ductal Adenocarcinoma (PDAC) Models"

_ijms, 2024, doi:10.3390/ijms25021265_

Round 1

Reviewer 1 Report

Comments and Suggestions for Authors

Interesting approach to use small molecule inhibitors targeting epigenetic molecules to enhances oncolytic HAdVC5 replication and efficacy in PDAC models.

Comments:

1.) Please specificy in more detail, why this small molecule library (181 compounds) still was useful to identify molecules that increase viral replication especially in PDAC models?

2.) Enhanced cell killing was also detected in Suit-2 and Panc04.03 cells infected with both oncolytic mutants in the presence of the inhibitors (Supplementary Fig. 4). This seems not to be the case for the Ad3delta-A20T construct?

3.) Labeling Figure 3c is of poor quality and should be improved. What is the x-axis? (days?)

4.) Line 280: rather use "indicate" insteadt of "prove"

5.) Line 304: replication in vivo: Please comment, why Neo-2734 was chosen for these experiments and whether the two other compounds (OTX-015 and iBet-762) did not show effects or the experiments were not done?

6.) Writing of inhibitors is not consistent? e.g.  iBet-762 vs. IBET762 and OTX-015 vs. OTX015

7.) Figure 6 is of poor quality. Figure 6C-E should be explained better in the legend.

8.) Line 465: What is meant with CAR-expression?

9.) Discussion from line 476 to 510: There are a lot suggestions how BRD4 inhibitors act to increase cell lysis and viral replication. A summary cartoon integrating the main ideas would be helpful.

10.) As epigenetic regulation of viral regulation by BRD inhibitors is a key finding of this manuscript, could you shortly comment whether DNA methylation inhibitors could be effective as well?

Author Response

Comments and Suggestions for Authors

Interesting approach to use small molecule inhibitors targeting epigenetic molecules to enhances oncolytic HAdVC5 replication and efficacy in PDAC models.

Comments: First we would like to thank the reviewer for the valuable comments and hope that we have made the requested changes and answered the specific comments below. 

1) Please specificy in more detail, why this small molecule library (181 compounds) still was useful to identify molecules that increase viral replication especially in PDAC models?

We have now added more information in the material and methods section to explain more details. This is an Epigenetics Compound Library with a unique collection of 181 molecules enriched in epigenetics compounds (Cat L1900, Selleckchem, USA) and was used as the inhibitor source. These molecules have been used in several solid tumour indications including PDAC.

2.) Enhanced cell killing was also detected in Suit-2 and Panc04.03 cells infected with both oncolytic mutants in the presence of the inhibitors (Supplementary Fig. 4). This seems not to be the case for the Ad3delta-A20T construct?

The Ad-3∆-A20T mediated cell killing was enhanced with iBET-762 and OTX-015 in Suit-2 cells and with iBET-762 and Neo-2734 in Panc04.03 cells. Ad∆∆ mediated cell killing was enhanced by all three inhibitors in both cell lines.

3.) Labeling Figure 3c is of poor quality and should be improved. What is the x-axis? (days?)

Thanks for pointing this out, our mistake. We have now corrected this and think the figure is more understandable.

4.) Line 280: rather use "indicate" insteadt of "prove". Thanks, this has been changed to: indicate.

5.) Line 304: replication in vivo: Please comment, why Neo-2734 was chosen for these experiments and whether the two other compounds (OTX-015 and iBet-762) did not show effects or the experiments were not done?

For the in vivo studies we only used Neo-2734 since this is the newest and reportedly more potent compound. We also wanted to keep the in vivo studies to a minimum because of cost, time and limit the use of animals. Furthermore, we found in our in vitro studies that all three inhibitors interacted with virus in a similar fashion and anticipated that Neo-2734 would be the best compound for potential future clinical applications. We have now tried to make this more clear in the text Line 254-256: ‘In these studies we selected Neo-2734 as the representative inhibitor since similar activities and efficacy in vitro was observed in combination with all three inhibitors.’

6.) Writing of inhibitors is not consistent? e.g.  iBet-762 vs. IBET762 and OTX-015 vs. OTX015.

Thanks for highlighting this. We have now corrected this throughout the text to iBET-762, OTX-015 and Neo-2734. In most figures we have shortened the names due to space.

7.) Figure 6 is of poor quality. Figure 6C-E should be explained better in the legend.

We have now improved the figure and added more explanations in the figure legend. ‘B) Venn diagram illustrating the interactions of iBet-762, virus, and the combined treatment, red and blue numbers indicate downregulated and upregulated genes respectively. C-E) Data analysed by GSEA using Broad Institute Hallmark gene sets (39). Normalised enrichment scores (NES) and Benjamini-Hochberg adjusted p-values are shown. Positive NES indicate upregulated pathways while negative scores denote that the pathway is downregulated by treatment.’

8.) Line 465: What is meant with CAR-expression?  Coxsackievirus and Adenovirus Receptor (CAR), that was described in the introduction Line 64.

9.) Discussion from line 476 to 510: There are a lot suggestions how BRD4 inhibitors act to increase cell lysis and viral replication. A summary cartoon integrating the main ideas would be helpful. 

Thanks for the suggestion, we have carefully considered this and understand that it may facilitate the understanding for the reader. However, we have decided not to attempt an illustration of pathway interactions since further investigations are needed to elucidate the exact order of events. These interactions are highly complex with numerous viral and cellular proteins and we do not want to present facts that are not yet completely understood and mislead the scientific community. We hope we can further explore these pathways in future studies.

10.) As epigenetic regulation of viral regulation by BRD inhibitors is a key finding of this manuscript, could you shortly comment whether DNA methylation inhibitors could be effective as well?

Methyltransferase inhibitors were included in the library for example, Decitabine and SGI-1047 but had no effect on viral replication in our study. CPI-360, a histone methyltransferase inhibitor had minor effects on viral replication (Fig. 1E). To our knowledge we are not aware of any clear reports suggesting that DNA methylation patterns affect the viral life cycle although plausible.

Submission Date

05 November 202

Date of this review

13 Dec 2023 12:37:09

Reviewer 2 Report

Comments and Suggestions for Authors

Oncolytic viruses are promising antitumor agents. In this work, authors consider the possibility of increasing the therapeutic effect when oncolytic adenoviruses and chemical inhibitors of bromodomain proteins are used together.  The developed approach is proposed to be used for the treatment of pancreatic adenocarcinoma. Authors screened a small molecule compound library (181 molecules) to identify molecules that increased viral replication in PDAC models. The most effective compounds have been used in in vivo experiments. The work is performed at a high level, has an unconditional scientific value. Nevertheless, for publication it is necessary to carry out major revision and answer the following questions:

1)      Please decipher HAdVs when it is mentioned first time in the abstract .

2)      What do you mean in the Abstract: “reverse drug resistance” or kill drug resistant tumors?

3)      Please, describe term “bromodomain” in the Abstract and in Introduction

4)      Please, describe (in the Introduction) how does adenoviruses entre the cells (in the context of specificity)?

5)      Please, describe why Suit-2 cell line was used as a model for molecular screening?  

6)      Please, please sign the data (top and bottom rows) on Fig. 1B.  Please explain why data with different background fluorescent signal values are presented? Why the figure caption does not indicate which groups are being compared (*) and what level of significance it corresponds to?

7)      Singular or average values are presented In Fig. 2A, B ?

8)      In Fig. 2 D, E, the axis signatures should be adequately presented (size of the signatures and meaning of the signatures). What do the numbers above and below the curves mean? % and molarity (concentration) should be better indicated.

9)      The captions on the left side of Figure 3A are not readable. Explain why flow cytometry was used for the analysis of intracellular protein Ki67? Is the % of Ki67-positive cells an average value presented or an individual value?

10)   In Figure 3B, please label the MW of the observed proteins. Why is actin used for normalization of nuclear proteins and not a specific nuclear protein? Please provide data (can be in Suppl.) on how samples with nuclear and cytoplasmic fractions looked when stained with Coomassie blue (or other protein dyes) to assess the quality of lysates of individual fractions.

11)   In Fig. 3C please make either color differentiation of separate groups or add hatching of lines (if you want to keep the black and white version of the figure). Otherwise it is difficult to understand the data.

12)   The quality of captions text on Figure 6 is poor. Please correct the caption of Fig. 6B: " B) Venn diagram illustrating the interactions of iBET-762, virus, and the combined treatment. individual inhibitors with virus on gene regulation".

Author Response

Oncolytic viruses are promising antitumor agents. In this work, authors consider the possibility of increasing the therapeutic effect when oncolytic adenoviruses and chemical inhibitors of bromodomain proteins are used together.  The developed approach is proposed to be used for the treatment of pancreatic adenocarcinoma. Authors screened a small molecule compound library (181 molecules) to identify molecules that increased viral replication in PDAC models. The most effective compounds have been used in in vivo experiments. The work is performed at a high level, has an unconditional scientific value. Nevertheless, for publication it is necessary to carry out major revision and answer the following questions:

We want to thank the reviewer for the insightful and detailed feedback and hope that we have included all the requested changes and responded to all the comments below.

1)      Please decipher HAdVs when it is mentioned first time in the abstract. Sorry, our error, now corrected: (Human AdenoViruses (HAdVs) (Line 16-17)

2)      What do you mean in the Abstract: “reverse drug resistance” or kill drug resistant tumors?

We have now tried to explain this better. Adenoviruses can sensitize drug resistant cancer cells so that the drug and virus combination act synergistically to kill even drug resistant cells: ‘Several oncolytic (Human AdenoViruses (HAdVs) have been reported to re-sensitize drug resistant cancer cells and in combination with chemotherapeutics attenuate solid tumour growth.’

3)      Please, describe term “bromodomain” in the Abstract and in Introduction.

The following has now been added to the introduction Line 91-98: ‘Three highly potent bromodomain-4 (BRD4) and p300/CBP inhibitors, OTX-015, iBet-762 and Neo-2734 were selected for further in-depth studies. Bromodomains (~110 amino acids) are present in numerous regulatory proteins and are essential for association with acetylated lysine residues on N-terminal histone tails. This binding mediates chromatin remodeling in turn facilitating active gene transcription. Members of the bromodomain containing protein family are the Bromo- and Extra-Terminal domain (BET) proteins including BRD4 that is part of the transcription complex regulating cMyc expression.’ We also added a few details in the abstract: Line 21’… an enriched epigenetics small molecule library…’ and Lines 23-24:’… three epigenetic inhibitors targeting bromodomain (BRD) containing proteins. Specifically, BRD4 inhibitors …’

4)      Please, describe (in the Introduction) how does adenoviruses entre the cells (in the context of specificity)?

We have now added the following description to explain the specificity Line 63-70: Ad∆∆ infects epithelial cells including adenocarcinomas by binding of the viral fibre protein to the Coxsackievirus and Adenovirus Receptor (CAR) followed by penton-binding to avß3- and avß5-integrins. The virus is subsequently internalized via the endosome, and the viral DNA transported to the nucleus for gene transcription. The Ad-3∆-A20T mutant has an additional alteration to improve bioavailability and decrease uptake in normal epithelial cells. Ad-3∆-A20T is de-targeted from CAR-binding and re-targeted to avß6-integrin expressing tumours by insertion of a 20-amino acid avß6-integrin-ligand from Foot-and-Mouth-Disease-Virus (FMDV) in the fibre knob (14, 21).

5)      Please, describe why Suit-2 cell line was used as a model for molecular screening?  

We explained in the result section that the Suit-2 cell line was used as a representative PDAC cell line. We have now added the following, Line 111-115: Culture conditions for the PDAC-representative Suit-2 cells, expressing the typical activating KRAS mutation, CDKN2A/p16 deletion and inactivating TP53 mutations [Jones 2008] were optimised. The growth characteristics for Suit-2 cells enabled growth in 384-well format and infection with Ad5wtGFP at increasing doses and time.

6)      Please, please sign the data (top and bottom rows) on Fig. 1B.  Left panel)

This is a representative 384-well plate and the edges of the plate show up as grey squares that do not include wells. The concentration range of virus has also been added in the legend (..increasing viral doses (125-4000 particles/cell (ppc). The middle panel has now been described in the figure legend; the wells in the bottom row are magnifications of the well in the top row to illustrate experimental readouts.    

Please explain why data with different background fluorescent signal values are presented?

The green is GFP expressed from the virus only in the virus infected cells after viral gene expression as described in the legend. The red fluorescence is the Hoechst dye that is only expressed in live cells. The merged images show live virus-infected cells, both green and red, that were used for detection of increased replication.

Why the figure caption does not indicate which groups are being compared (*) and what level of significance it corresponds to?

Thanks for highlighting this omission, the p-values have now been included in the legend: *p<0.05, ***p<0.001

7)      Singular or average values are presented In Fig. 2A, B ?

We have now clarified this by adding the following in the legend to Fig 2A-B: ‘…. averages triplicate samples, one representative experiment.’

8)      In Fig. 2 D, E, the axis signatures should be adequately presented (size of the signatures and meaning of the signatures). What do the numbers above and below the curves mean? % and molarity (concentration) should be better indicated.

We have now improved these figures; x-axes are virus dose in particles per cell (ppc), y-axes is cell death in % (compared to total live cells), the indicated drugs were included at the indicated concentrations in µM (above the respective graphs). The numbers above and below the graphs are the respective virus and virus + drug EC50-values in ppc.

9)      The captions on the left side of Figure 3A are not readable. Explain why flow cytometry was used for the analysis of intracellular protein Ki67? Is the % of Ki67-positive cells an average value presented or an individual value?

We have now improved the text on the graph in Fig 3A. The y-axis indicates Ki67 positive cell counts and are the results from a regular and representative flowcytometry study of fluorescent intensity and % positive (samples in duplicates). Flow cytometry after cell membrane permeation and protein fixation is a standard method to quantify Ki67. The alternative is immunblotting that is less quantitatively reliable. In Fig 3A right panel each sample was measured in triplicates and data from one experiment over time is presented.

10)   In Figure 3B, please label the MW of the observed proteins. Why is actin used for normalization of nuclear proteins and not a specific nuclear protein? Please provide data (can be in Suppl.) on how samples with nuclear and cytoplasmic fractions looked when stained with Coomassie blue (or other protein dyes) to assess the quality of lysates of individual fractions.

The molecular weights have now been included; c-Myc 57-65kDa, E1A 26-53kDa and Actin 42kDa. Actin was used to normalize both cytoplasmic and nuclear proteins to enable quantitative comparisons of c-Myc and E1A in each fraction. c-Myc is the specific nuclear marker and was not detected in the cytoplasmic preparations. We did not preserve blots stained for total proteins since we were only interested in determining the effects of the inhibitors on the expression levels of the viral E1A protein and its interactions with c-Myc expression. The distribution pattern of E1A in the nuclear and cytoplasmic fractions are in agreement with previous findings in our laboratory.

11)   In Fig. 3C please make either color differentiation of separate groups or add hatching of lines (if you want to keep the black and white version of the figure). Otherwise it is difficult to understand the data.

Thanks for pointing this out. We have now changed the figures to include colored lines and better descriptions of axes, units and treatments.

12)   The quality of captions text on Figure 6 is poor. Please correct the caption of Fig. 6B: " B) Venn diagram illustrating the interactions of iBET-762, virus, and the combined treatment. individual inhibitors with virus on gene regulation".

We have now added the following: ‘B) Venn diagram illustrating the interactions of iBet-762, virus, and the combined treatment, red and blue numbers indicate downregulated and upregulated genes, respectively.’

Submission Date

05 November 2023

Date of this review

24 Nov 2023 08:48:43

Reviewer 3 Report

Comments and Suggestions for Authors

In the manuscript entitled “Inhibition of bromodomain proteins enhances oncolytic HAdVC5 replication and efficacy in pancreatic ductal adenocarcinoma (PDAC) model”, the authors used several approaches to support the hypothesis that in response to 3 different drugs, oncolytic viruses had its replication enhanced, improving their efficacy against PDAC, in vivo and in vitro.

Even though the results are interesting and new, some issues raised the following concerns:

Major concerns:

1-      Concerning figure legend descriptions and image quality:

a.        In figure 2 legend, it is not clear the number of wells tested for each treatment. The same for figure 2B and figure 3A.

b.       The indication “right” or “left panel is lacking in the legend, leaving the second panel without reference on the legend (For figures 2A, 2B and 3A).

c.       In figure 2A, right panel, has the number of the y-axis unevenly positioned. Was this figure made using Prisma software? Please correct. In this very figure the left panel shows the flow cytometry histogram for Ki67, but the legends for the colors are illegible. In this very figure, x-axis, the logarithm scale is blurred and uneven, mainly between -103 to 103. Please provide a better representative figure for the result.

d.       In figure 2-C, right panel, the y-axis scale is written as 3e4. Please correct using proper notation.

e.       In figure 3A, right panel, the y-axis scale possesses different intensity color, what could be interpreted as improper image correction.

f.        In figure 4A, lower panel, the y-axis scale is off. With this (e4). was It generated automatically by the software? Could be better to place in the legend that the numbers of the scale are x104.

g.       Please revise scales for x and y-axis for all figures, so as the descriptions in the figure legends.

 The conclusions are supported by the data, but the figures and legend descriptions must be improved. Thus, this review RECOMMENDS  the publication  of the manuscript AFTER MANDATORY CORRECTIONS IN FIGURES AND FIGURE LEGENDS.

Author Response

In the manuscript entitled “Inhibition of bromodomain proteins enhances oncolytic HAdVC5 replication and efficacy in pancreatic ductal adenocarcinoma (PDAC) model”, the authors used several approaches to support the hypothesis that in response to 3 different drugs, oncolytic viruses had its replication enhanced, improving their efficacy against PDAC, in vivo and in vitro.

Even though the results are interesting and new, some issues raised the following concerns:

We want to thank the reviewer for the insightful and detailed feedback and hope that we have included all the requested changes and responded to all the comments below.

Major concerns:

  • Concerning figure legend descriptions and image quality:
  1. In figure 2 legend, it is not clear the number of wells tested for each treatment. The same for figure 2B and figure 3A. Thanks for highlighting this omission. We have now corrected this by adding the following in the legend to Fig 2A-B: ‘…. averages, triplicate samples, one representative experiment.’ For Fig. 3A we clarified this as follows: ‘ A) Suit-2 cells were treated with DMSO (ctrl), iBET-762 (0.5µM) or OTX-015 (0.2µM) for 24, 48 and 72h followed by Ki67 staining and flow cytometry. Left panel) flow cytometry profile from one representative experiment, (right panel) quantification of Ki67 flow cytometry data over time with increasing concentrations of drugs (representative data, duplicate wells).’
  2. b.       The indication “right” or “left panel is lacking in the legend, leaving the second panel without reference on the legend (For figures 2A, 2B and 3A). For Fig 2A and B, the following is already described in the legend: A-B) Suit-2 cells were infected with AdwtGFP at 1pfu/cell and treated with the inhibitors at the indicated doses and analysed by flow cytometry after 24h (A) and from 24-72h (B). (Left panels) percentage of GFP positive cells, (Right panels) mean fluorescence intensity. Polybrene at 6µg/ml was used as positive control for adenovirus transduction, averages triplicate samples, one representative experiment. For Fig 3A left and right have now been added: A) Suit-2 cells were treated with DMSO (ctrl), iBET-762 (0.5µM) or OTX-015 (0.2µM) for 24, 48 and 72h followed by Ki67 staining and flow cytometry. Left panel) flow cytometry profile from one representative experiment, (right panel) quantification of Ki67 flow cytometry data over time with increasing concentrations of drugs (representative data, duplicate wells).
  3. In figure 2A, right panel, has the number of the y-axis unevenly positioned. Was this figure made using Prisma software? Please correct. In this very figure (the reviewer must mean Fig. 3A?) Thanks for pointing this out, this is now corrected. the left panel shows the flow cytometry histogram for Ki67, but the legends for the colors are illegible. In this very figure, x-axis, the logarithm scale is blurred and uneven, mainly between -103to 103. Please provide a better representative figure for the result. We have now corrected this and hope it is more clear.
  4. In figure 2-C, right panel, the y-axis scale is written as 3e4. Please correct using proper notation. Thanks for noticing, we have now changed this to 10000-30000.
  5. In figure 3A, right panel, the y-axis scale possesses different intensity color, what could be interpreted as improper image correction. Not sure what the reviewer suggests? The y-axis is from 0-110% Ki67 positive cells (based on total cell numbers). The bars in the graph have different colors to facilitate understanding; black bars are no treatment or DMSO only, grey bars are treatments with 0.5µM iBet-762 and white bars are treatment with 0.2µM OTX-015.
  6. In figure 4A, lower panel, the y-axis scale is off. With this (e4). was It generated automatically by the software? Could be better to place in the legend that the numbers of the scale are x104. Thanks for noticing this. We have corrected the figure and the y-axis is now labelled from 0-7 X10000, hope this is more clear.

 The conclusions are supported by the data, but the figures and legend descriptions must be improved. Thus, this review RECOMMENDS  the publication  of the manuscript AFTER MANDATORY CORRECTIONS IN FIGURES AND FIGURE LEGENDS.

Submission Date

05 November 2023

Date of this review

Round 2

Reviewer 1 Report

Comments and Suggestions for Authors

The authors have addressed my comments satisfactorily. However, while reading the revision, I noticed one point that I, as a molecular diagnostician, cannot let stand:

Line 114: Culture conditions for the PDAC-representative Suit-2 cells, expressing the typical activating KRAS mutation, CDKN2A/p16 deletion and inactivating TP53 mutations (29),

The is no typical KRAS mutation, as mutations may occur in codon 12, 13, 59 and 61 (hotspots). In pancreatic cancer, KRAS G12D mutations are most common. Is this mutation meant?

Author Response

We thank the reviewer for helping us to make the manuscript more clear.

We have now added the specific mutation of Kras in the Suit-2 cells: 

Line 113-114: Suit-2 cells, expressing the typical activating KRAS mutation (KrasG12D), CDKN2A/p16 deletion and inactivating TP53 mutations (29), ....

Reviewer 2 Report

Comments and Suggestions for Authors

The authors have significantly improved the manuscript and it can be published in its current form. I have only 2 minor comments:

1)      L. 16: “Several oncolytic (Human Adeno Viruses (HAdVs)” – please, delete an extra bracket.

2)      I suggest that the authors change the abbreviation CAR (Coxsackievirus and Adenovirus Receptor), since in oncology this designation refers specifically to the chimeric antigen receptor.

Author Response

We thank the reviewer for taking the time to read the manuscript and feedback to us. We have now added the following changes: 

1)      L. 16: “Several oncolytic (Human Adeno Viruses (HAdVs)” – please, delete an extra bracket. Thanks, bracket has now been removed.

2)      I suggest that the authors change the abbreviation CAR (Coxsackievirus and Adenovirus Receptor), since in oncology this designation refers specifically to the chimeric antigen receptor. Unfortunately this is the accepted abbreviation for the receptor and we cannot change this since it has been used in viral literature for decades. However, to make it more clear in addition to the explanation on Line 64 in the introduction, we also added the following in the discussion, Line 489: '....... without affecting the Coxsackievirus and Adenovirus Receptor (CAR)-expression......'

Reviewer 3 Report

Comments and Suggestions for Authors

The revised version of the manuscript now meets the quality standards of the journal. Thus, this reviewer RECOMMENDS the publication in IJMS.

Author Response

We want to thank the reviewer for the extensive reading and the valuable feedback on our manuscript. Much appreciated.